# Real-World Experience of Monitoring Practice of Endocrinopathies Associated with the Use of Novel Targeted Therapies among Patients with Solid Tumors

**DOI:** 10.3390/medsci10040065

**Published:** 2022-11-21

**Authors:** Atika AlHarbi, Majed Alshamrani, Mansoor Khan, Abdelmajid Alnatsheh, Mohammed Aseeri

**Affiliations:** 1Department of Pharmaceutical Care Services, Princess Noorah Oncology Center (PNOC), King Abdulaziz Medical City (KAMC), Ministry of National Guard Health Affairs (MNGHA), P.O. Box 9515, Jeddah 21423, Saudi Arabia; 2King Abdullah International Medical Research Center’s (KAIMRC), Riyadh 11481, Saudi Arabia

**Keywords:** targeted therapy, monitoring errors, endocrinopathies, thyroid disorders, hyperglycemia, dyslipidemia

## Abstract

Background: Cancer treatments have gradually evolved into targeted molecular therapies characterized by a unique mechanism of action instead of non-specific cytotoxic chemotherapies. However, they have unique safety concerns. For instance, endocrinopathies, which are defined as unfavorable metabolic alterations including thyroid disorders, hyperglycemia, dyslipidemia, and adrenal insufficiency necessitate additional monitoring. The aim of this study was to assess the prevalence of monitoring errors and develop strategies for monitoring cancer patients who receive targeted therapies. Method: A retrospective chart review was used to assess the prevalence of monitoring errors of endocrinopathies among cancer patients who received targeted therapies over one year. All of the adult cancer patients diagnosed with a solid tumor who received targeted therapies were included. The primary outcome was to determine the prevalence of monitoring errors of endocrinopathies. The secondary outcomes were to assess the incidences of endocrinopathies and referral practice to endocrinology services. Results: A total of 128 adult patients with solid tumors were involved. The primary outcome revealed a total of 148 monitoring errors of endocrinopathies. Monitoring errors of the lipid profile and thyroid functions were the most common error types in 94% and 92.6% of the patients treated with novel targeted therapies, respectively. Subsequently, 57% of the monitoring errors in the blood glucose measures were identified. Targeted therapies caused 63 events of endocrinopathies, hyperglycemia in 32% of the patients, thyroid disorders in 15.6% of them and dyslipidemia in 1.5% of the patients. Conclusion: Our study showed a high prevalence of monitoring errors among the cancer patients who received targeted therapies which led to endocrinopathies. It emphasizes the importance of adhering to monitoring strategies and following up on the appropriate referral process.

## 1. Introduction

In the last decade, the robust development of cancer treatments has been achieved, with the management of cancer moving more towards the use of targeted molecular therapies instead of non-specific cytotoxic chemotherapies [1]. The targeted therapies have changed the paradigm of cancer treatment as these are characterized by a unique mechanism of action which is considered to be highly specific for the key cellular biological pathways implicated in the cancer process [2]. Notwithstanding their effectiveness, these new, targeted molecular therapies exhibit certain side effects such as endocrinopathies which have a significant impact on the patients’ quality of life in the long run [1].

Endocrinopathies are among the most common side effects associated with the use of immune checkpoint inhibitors and targeted therapies (refer to Appendix A) [3]. Endocrinopathies are defined as unfavorable metabolic alterations which include hyperthyroidism, hypothyroidism, hyperglycemia, hypertriglyceridemia, hypercholesterolemia, hypogonadism and adrenal insufficiency. The incidence of endocrinopathies with the use of immune checkpoint-blocking antibodies and targeted therapies has been difficult to accurately state due to the varied methods of assessment, diagnosis and monitoring in different clinical trials [4]. The most commonly reported endocrinopathies are hypothyroidism, hyperthyroidism, hyperglycemia, dyslipidemia and hypophysitis [5,6,7,8,9]. As a result of numerous endocrinopathies being reported in real-life practice, the European Society for Medical Oncology (ESMO), the American Society of Clinical Oncology (ASCO) and the National Comprehensive Cancer Network (NCCN) have established clinical practice guidelines for the diagnosis and management of immunotherapy-related toxicities [10,11,12]. On the other hand, targeted therapies which include small molecule tyrosine-kinase inhibitors (TKIs), the mammalian target of rapamycin (mTOR) inhibitor, immunomodulatory agents and other monoclonal antibodies have had no established guidelines for clinical practice in the diagnosis, monitoring and management of their related toxicities apart from the recommendations that are mentioned in their prescribing information. As many studies have shown, novel targeted therapies have durable clinical benefits which highlight the importance of establishing safe standards in preventing endocrinopathies that are associated with the use of novel targeted therapies [3,4,5,6,7,8,9].

Hence, novel targeted therapies require additional monitoring and an appropriate clinical review [13]. The failure to review a prescribed targeted therapy is defined as a monitoring error. The American Society of Health System Pharmacists (ASHP) defined a monitoring error as a failure to review a prescribed regimen for the appropriateness and detection of problems, or the failure to use appropriate clinical or laboratory data for an adequate assessment of the patient’s response to prescribed therapy [14].

Di Lorenzo et al. published a systematic analysis of the abstracts issued by ASCO and ESMO between 2005 and 2010. Their aim was to describe the side effects associated with targeted therapy that are used to treat metastatic renal cell carcinoma (mRCC) and its management. Furthermore, they investigated the incidence and grading of the toxicities associated with certain targeted therapies which are used in the management of mRCC, and they concluded that targeted therapies caused severe toxicities, thereby requiring external specialist consultation and treatment modification [15]. Another review article was published in 2013 by Grace K et al. with the aim of providing a brief overview of the toxicities associated with novel agents and relevant implications for the management of their side effects in cancer patients. They found that targeted therapy agents rose to unanticipated toxicities despite their efficacy profile due to previously unknown mechanisms and/or the multiplicity of the affected off-target proteins [16]. Moreover, many studies have evaluated the incidence and prevalence of targeted therapy–related toxicity, and they have recommended effective strategies to prevent these safety hazards [17,18].

The growing evidence of a significant number of incidences of endocrinopathies associated with the use of novel targeted agents means that healthcare providers ought to be aware of these side effects and know how to monitor them periodically. 

At our institution, multiple targeted therapies are used for a wide variety of oncology indications. These medications can cause many types of endocrinopathies that need close monitoring and regular follow-ups in cancer patients. Unfortunately, there is no standard practice for the safe monitoring of these medications to ensure patient safety, especially since this medications usually given in an outpatient setting. The aim of our study was to assess the prevalence of monitoring errors of endocrinopathies and then, develop strategies for safe practices in the monitoring of patients who are undergoing targeted therapies.

## 2. Methods

### 2.1. Study Design

This retrospective cohort study evaluated the prevalence of monitoring errors of endocrinopathies in patients receiving novel targeted therapies who were included in our hospital formulary for the treatment of adult patients with solid tumors at the National Guard of Health Affairs—King Abdulaziz Medical City—in the western region (KAMC-WR), Saudi Arabia, during the period from 1 June 2016 to 31 December 2017. The Investigational Review Board’s (IRB) approval was obtained in February 2018.

### 2.2. Study Population 

All of the adult patients who were 18 years old or older, had been diagnosed with a solid tumor, and received one of the novel targeted therapies that is available in our formulary (i.e., nivolumab, atezolizumab, everolimus, sorafenib, sunitinib, pazopanib, regorafenib, or abiraterone) were eligible for the study. The patients treated with novel targeted therapies for hematological malignancies were excluded from the study due to limited time allocated for the study. 

### 2.3. Study Procedures

The patients were identified through the Information System Development (ISD) department, and the data were retrieved through the patients’ electronic medical records. We reviewed the electronic medical records using hospital information systems to obtain the pertinent laboratory and medical data. For each patient, we recorded the following variables: demographic characteristics (age, gender, diagnosis, date of diagnosis and comorbidities) and the presence of previously reported adverse drug events (ADEs) that were secondary to novel targeted therapies treatment. Furthermore, the percentage of monitoring errors associated with the use of novel targeted therapies among the cancer patients occurred at the baseline and during the follow-up visits, and they were defined as a failure to use appropriate clinical or laboratory data (as fasting blood glucose, random blood glucose, HbA1c, T_3_, T_4_, Thyroid-stimulating hormone (TSH) and lipid profiles (LDL, triglyceride and total cholesterol) for the adequate assessment and appropriate monitoring of the patient’s response to a prescribed therapy as per recommended. These monitoring errors were identified by using the Lexicomp program (online commercial drug information database) and the Food and Drug Administration (FDA) who disseminate information for each medication (refer to Appendix A). Moreover, we tracked the patients who received the novel targeted therapies at our institution and had any endocrinopathies based on the clinician assessment and the laboratory results. Then, we assessed the percentage of referrals to the endocrinologist for the indicated patients with endocrinopathies. The data were collected and analyzed over a period of 6 months. 

### 2.4. Statistical Analysis

The Statistical Package for the Social Sciences (SPSS) version 24 was used for data analysis. The descriptive statistics are shown as the means (95% confidence intervals, CI), frequencies or percentages when it is appropriate. The data are organized and summarized in the tables.

## 3. Results 

A total of 128 adult patients diagnosed with solid tumors were included in this study. The baseline characteristics of the participants are shown in Table 1. Among 128 oncology patients, 14 (11%) patients received immune checkpoint inhibitors (nivolumab, atezolizumab), while 114 (89%) patients were on TKIs, mTOR and abiraterone. We found the inadequate monitoring of thyroid function and blood glucose in thirteen (92.8%) and two (14.2%) patients on the immune checkpoint inhibitors, respectively, as they required thyroid function tests and blood glucose test at the baseline and periodically (Table 2). While 50 (92.6%) patients on TKIs and 52 (64.2%) patients on sunitinib, everolimus and abiraterone had inadequate monitoring of their thyroid function and blood glucose levels, respectively, as they required thyroid function tests and blood glucose tests at the baseline and periodically. We found that up to 31 (94%) patients had monitoring errors associated with the use of oral everolimus which required a lipid profile test at the baseline and periodically (Table 3). Furthermore, we found that novel targeted therapy had caused 63 incidents of endocrinopathies among adult solid tumor patients, including hyperglycemia in 32% (41) of the patients, hypothyroidism and/or hyperthyroidism in 15.6% (20) of the patients and dyslipidemia in 1.5% (2) of the patients. A total of 11 patients were lost in their follow-up due to their referral to other oncology center or disease progression. Moreover, we assessed the referral to the endocrinologist for the indicated patients, and found that nearly 27% (17 patients from a total of 63 patients) of those who had endocrinopathies were referred to an endocrine specialist (Table 4).

## 4. Discussion

Despite the efficacy of the new, targeted molecular therapies, several studies have highlighted their side effects such as endocrinopathies [3,4,5,6,7,8,9]. Endocrinopathies are among the most common side effects associated with the use of immune checkpoint inhibitors and targeted therapies, which have a major impact on the patients’ quality of life in the long term [5,6,7,8,9]. Establishing clinical practice guidelines for the diagnosis and management of immunotherapy-related toxicities is essential to ensuring the patients’ safety and possible efficacy [10,11,12]. Our study has shown a high incidence of endocrinopathies among the cancer patients who received a novel targeted therapy, which has been difficult to state precisely due to the variable methods of assessment, diagnosis and monitoring. The most common endocrinopathies are hyperglycemia, hypothyroidism and hyperthyroidism. These endocrinopathies, which are caused by novel targeted therapies, can be prevented through proper monitoring and early detection, as it is supported in many studies [19,20,21,22,23,24].

A delay in identifying the endocrinopathies associated with targeted molecular therapies can limit the opportunities to provide supportive care to minimize or adequately manage these side effects. Hence, any failure to review a prescribed regimen for appropriateness or to use appropriate clinical or laboratory data for an adequate assessment of the patient’s response or safety could lead to a monitoring error [14]. In 2016, Marabelle and his colleagues came up with some collaborative institutional guidelines that emphasize the five pillars of immunotherapy toxicity management. These pillars include how to prevent, anticipate, detect, treat and monitor immunotherapy-related toxicity [25]. A systematic review of ten and thirty-eight randomized control trials published in 2014 and 2018, respectively, demonstrated that the use of immune checkpoint inhibitors and TKIs are associated with an increased risk of hypothyroidism, hyperthyroidism, adrenal insufficiency and hypophysitis [26,27].

Our study found the inadequate monitoring of the thyroid functions in patients who are on novel targeted medication (i.e., nivolumab, atezolizumab, sorafenib, sunitinib, pazopanib and regorafenib). We also found that 94% of the study patients had an inadequate monitoring of their lipid profiles whilst they were on everolimus, while 57% of the patients who were on novel targeted medications (i.e., nivolumab, atezolizumab, everolimus, sunitinib and abiraterone) had monitoring errors in their blood glucose levels.

Moreover, when we assessed the referral to the endocrinologist for the indicated patients, we found that only 27% of the affected patients were referred which is considered to be a medical malpractice and inconsistent with the current experts’ recommendations [10,11,12,20,21,22,28]. Many studies recommend that oncologists should seek an endocrinologist specialist or internist support for two reasons: for oncologists to learn the proper management of specific endocrinopathies toxicities, and for the organ specialists to increase their knowledge about these new drug-related toxicities, thereby creating a virtuous circle for the patients’ management [25,29,30].

To the best of our knowledge, this is the first published study of its kind that investigated the monitoring practice of endocrinopathies associated with the use of novel targeted therapies. Additionally, due to this established association among targeted molecular therapies and endocrinopathies, even with a small number of patients, it is now recommended that these toxicities should be identified as they may easily be overlooked among many patients.

Our study has several limitations such as we only observed a single institution and we included novel targeted therapies that were available during the study period so we were unable to assess if the patients had access to different healthcare systems in order to manage their chronic diseases which could contribute to the accuracy of the study results. Moreover, 11 patients were lost in their follow-up due to their referral to other oncology centers or disease progression, and as such, they could not be included to determine the true incidence of endocrinopathies, and this might have led to failure in monitoring. The sample size was not calculated as we include all of the cancer patients who received the selected medications for solid tumor indications at our center during the period from 1 June 2016 to 31 December 2017. Despite these limitations, the study demonstrated a high prevalence of monitoring errors associated with the use of targeted anticancer medications, and we identified the incidence of endocrinopathies that were secondary to novel targeted therapies among adult patients with solid tumors. The strengths of the study include the variety of sources that were used to identify the incidence of endocrinopathies. Rather than epidemiologically valid incidence estimates, these reports provide a descriptive account of a whole range of medication-related monitoring errors and facilitate the development of interventions to address the areas of risk and improve the patients’ quality of life.

## 5. Conclusions

Our study demonstrated a high prevalence of monitoring errors associated with the use of targeted anticancer medications and identified the incidence of endocrinopathies that were secondary to novel targeted therapies among adult patients with solid tumors. The causes of these errors have been identified as being multifaceted, involving various members of a multidisciplinary team. Our findings emphasize the importance of establishing monitoring guidelines for novel targeted therapies and the need for collaborative efforts among health care providers to optimize the cancer treatment outcomes. To that end, several strategies should be considered and implemented to overcome monitoring errors, as novel chemotherapy now constitutes a new era in cancer treatment.

## Figures and Tables

**Table 1 medsci-10-00065-t001:** Baseline Characteristics.

Baseline Characteristics (N = 128)	
Age [year] (Mean ± SD)	63 ± 14
Gender no. (%)	Male 78 (61%)
	Female 50 (39%)
Coexisting conditions no. (%)	None 31 (24%)
	One comorbidity 32 (25%)
	≥2 comorbidities 65 (51%)
Coexisting endocrine disorders no. (%)	- Diabetes Mellitus 47 (33.3%) - Dyslipidemia 10 (7%) - Thyroid disorders 10 (7%) - No coexisting endocrine 74 (52.7%)
Diagnosis no. (%)	- Metastatic prostate cancer 27 (21%) - Metastatic breast cancer 26 (20.3%) - Hepatocellular carcinoma 26 (20.3%) - Metastatic Renal cell carcinoma 24 (18.75%) - Neuroendocrine tumor 7 (5.5%) - Metastatic bladder cancer 3 (2.35%) - Refractory/Relapsed HL * 3 (2.35%) - Renal cell carcinoma 2 (1.56%) - Metastatic thyroid cancer 2 (1.56%) - Metastatic soft-tissue sarcoma 2 (1.56%) - Metastatic Gastrointestinal Stromal Tumor 2 (1.56%) - Metastatic colorectal cancer 2 (1.56%) - Uterine liposarcomas 1 (0.8%) - Non-small-cell lung carcinoma (NSCLC) 1 (0.8%)

* HL is treated under medical oncology (solid tumor team) at our institution.

**Table 2 medsci-10-00065-t002:** Prevalence of Monitoring Errors of Endocrinopathies in Patients Receiving Immune Checkpoint Inhibitors.

Targeted Medications (n)	Prevalence of Inadequate Monitoring Errors of Endocrinopathies in Patients Receiving Immune Checkpoint Inhibitors no. (%)	
	Monitoring of Blood Glucose	Monitoring of Thyroid Function Test
Atezolizumab (2)	2 (100%)	2 (100%)
Nivolumab (12)	0%	11 (91.6%)

**Table 3 medsci-10-00065-t003:** Prevalence of Monitoring Errors of Endocrinopathies in Patients Receiving Oral Targeted Chemotherapy.

Targeted Medications (n)	Prevalence of Inadequate Monitoring Errors of Endocrinopathies in Patients Receiving Oral Targeted Chemotherapy no. (%)		
	Monitoring of Blood Glucose	Monitoring of Thyroid Function Test	Monitoring of Lipid Profile Test
Abiraterone (27)	25 (93%)	NR	NR
Everolimus (33)	27 (82%)	NR	31 (94%)
Pazopanib (2)	NR	2 (100%)	NR
Regorafenib (2)	NR	2 (100%)	NR
Sorafenib (29)	NR	28 (97%)	NR
Sunitinib (21)	0%	18 (85.7%)	NR

NR: Not routinely recommended to monitor while receiving this medication.

**Table 4 medsci-10-00065-t004:** Incidents and Types of Endocrinopathies in Patients Receiving Targeted Chemotherapy.

Targeted Medications (n)	Incidence of Endocrinopathies in Patients Receiving Targeted Chemotherapy			Referral to Endocrinologist
	Altered Blood Glucose	Thyroid Disorders	Dyslipidemia	
Abiraterone (27)	12 patients had hyperglycemia	NR	NR	5 patients were referred
Atezolizumab (2)	1 patient had hyperglycemia	No identified incidence of endocrinopathies	NR	No referral
Everolimus (33)	16 patients had hyperglycemia	NR	2 patients had hypertriglyceridemia	6 patients were referred
Nivolumab (12)	4 patients had hyperglycemia	3 patients had hypothyroidism and 1 patient had hyperthyroidism	NR	2 patients were referred
Pazopanib (2)	NR	1 patient had hypothyroidism	NR	No referral
Regorafenib (2)	NR	No identified incidence of endocrinopathies	NR	-
Sorafenib (29)	NR	2 patients had hypothyroidism and 3 patients had hyperthyroidism	NR	2 patients were referred
Sunitinib (21)	8 patients had hyperglycemia	10 patients had hypothyroidism	NR	2 patients were referred
**Total incidence of endocrinopathies**	63 Incidence of Endocrinopathies			

NR: Not routinely recommended to monitor while receiving this medication. Eleven patients were lost in their follow-up (six of them were on abiraterone, two of them were on everolimus, two of them were on Sorafenib and one of them was on nivolumab).

## Data Availability

Not applicable.

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
