# Peer review of "Real-World Experience of Monitoring Practice of Endocrinopathies Associated with the Use of Novel Targeted Therapies among Patients with Solid Tumors"

_medsci, 2022, doi:10.3390/medsci10040065_

Round 1

Reviewer 1 Report

The manuscript by AlHarbi et al. assess the monitoring errors for cancer patients that received targeted therapies for over a year. The manuscript is well written and read easily. Below are specific comments and suggestions for the authors:

Introduction

·      The paragraph about the two reviews (Di Lorenzo and Grace) could be shortened a bit.

·      It is appreciated the authors end their introduction with the aim of the study.

Methods

·      TSH should be introduced before being abbreviated.

·      ® are redundant in scientific writing and should me omitted.

·      This sentence seems a bit out of place in the section about statistical analysis since that is the recruitment period and that is already stated above. Please remove the sentence:

o   “It was estimated that the study would include all adult patients diagnosed with solid 131 tumor and received one of the novel targeted therapies over the one year (June 1, 2016 to 132 December 31, 2017).”

·      The same goes for this sentence – it is already stated above what parameters are measured to assess endocrinopathies.

o   “Eligible patients were reviewed, and the required/stated parameters 133 were measured.”

Discussion

·      The first paragraph of the discussion is mostly a restatement of what is already known from the introduction, besides the sentences about the findings from the present study. Consider greatly reduce the paragraph to make it clearer and more concise.

·      What is meant by “different healthcare systems” in the following sentence? Please clarify.

o   “Our study have several limitations as we were unable to assess if patients had access 210 to different healthcare systems in order to manage their chronic diseases which could con- 211 tribute to the accuracy of the study results.”

·      In the limitation section, please specify the number of patients lost to follow up.

·      In the discussion it is stated “Marabelle and his colleagues recommended that oncologists should seek an endocrinologist specialist or internist support”. However, the aim of the present study was to “develop strategies for safe practices in the monitoring of patients on targeted therapies”. The authors supposed strategies to monitor needs to be discussed in greater and more specific details – what do the authors recommend should be amended or changed in the ready published guideline for this? The following statement from the discussion about this quite important question and aim of the study is very vague. Please elaborate extensively on this aspect.

o   “…it is now recommended that these toxicities should be identified as they may easily be overlooked 208 among many patients.”

·      References

o   The reference list is a bit short (only 23 references are cited). Consider adding more of the recent and most relevant studies about the topic. Also, some sentences require references to support their statements. There a several examples of that in the introduction and discussion.

Reviewer 2 Report

I wish to thank you for giving the opportunity to review this interesting manuscript.

The study presents a cohort of 128 patients with solid tumors treated with novel targeted therapies regarding monitoring endocrinopathies during treatment.

I have several remarks:

1.       Among the endocrinopathies presented in introduction is Adrenal insufficiency (line 15). In the section results it’s not mentioned at all, did the authors assess it?

2.       The results section should include how many patients were lost for follow-up.

3.       In the Methods section – study population, Hematologic malignancies were excluded (line 108-109). In Table 1, three patients have diagnosis of HL, why?

4.       Why lipid profile monitoring wasn’t assessed in Immune Check-point Inhibitors (Table 2).

5.       The total number of the patients mentioned in line 145-146 is higher than the total number in the study.

6.       I would recommend elaborating more on limitations (discussion section) - The study is single center study with small patient’s number. The study included only patients with solid tumors (There are other malignancies that are treated with these novel targeted therapies). The study didn’t assess all endocrinopathies (see comment #1 and #4).

Round 2

Reviewer 1 Report

My concerns have been addressed.

Author Response

We would like to thank the reviewer for the careful and thorough reading of our manuscript and for the thoughtful comments and constructive suggestions, which help to improve the quality of this manuscript.

Reviewer 2 Report

I would thank the authors for the reply's comments, I have several remarks:

Although the number of patients lost for follow-up is mentioned in line 238, I would recommend adding it to the results section.

I would recommend explaining in the results section/Table 1 why patients with HL were included in the study.

I would also recommend adding the number of errors for thyroid functions and lipid profile in Line 24,25, as readers might get confused with “thyroid functions were the most common error type” while there are 94% monitoring errors of lipid profile.

Author Response

Thanks for the prompot response, Please see the attachment.
